# An Alignment Method for Strapdown Inertial Navigation Systems Assisted by Doppler Radar on a Vehicle-Borne Moving Base

**DOI:** 10.3390/s19204577

**Published:** 2019-10-21

**Authors:** Bo Yang, Jianxiang Xi, Jian Yang, Liang Xue

**Affiliations:** Department of Control Engineering, Xi’an Research Institute of High Technology, Xi’an 710025, China; xijx07@mails.tsinghua.edu.cn (J.X.); xuelmems@163.com (L.X.)

**Keywords:** strapdown inertial navigation system, Doppler radar, dead reckoning, moving-base alignment, Kalman filter

## Abstract

In this study, we investigated a novel method for high-accuracy autonomous alignment of a strapdown inertial navigation system assisted by Doppler radar on a vehicle-borne moving base, which effectively avoids the measurement errors caused by wheel-slip or vehicle-sliding. Using the gyroscopes in a strapdown inertial navigation system and Doppler radar, we calculated the dead reckoning, analyzed the error sources of the dead reckoning system, and established an error model. Then the errors of the strapdown inertial navigation system and dead reckoning system were treated as the states. Besides velocity information, attitude information was cleverly introduced into the alignment measurement to improve alignment accuracy and reduce alignment time. Therefore, the first measurement was the difference between the output attitude and velocity of the strapdown inertial navigation system and the corresponding signals from the dead reckoning system. In order to further improve the alignment accuracy, more measurement information was introduced by using the vehicle motion constraint, that is, the velocity output projection of strapdown inertial navigation system along the transverse and vertical direction of the vehicle body was also used as the second measurement. Then the corresponding state and measurement equations were established, and the Kalman filter algorithm was used for assisted alignment filtering. The simulation results showed that, with a moving base, the misalignment angle estimation accuracy was better than 0.5’ in the east direction, 0.4’ in the north direction, and 3.2’ in the vertical direction.

## 1. Introduction

In order for vehicle-borne weapon systems to rapidly respond and maneuver, their inertial navigation systems are generally required to have rapid alignment capabilities. If high-accuracy alignment could be accomplished during vehicle motion, it would be of great military significance.

The initial alignment of a vehicle-borne inertial navigation system with a moving base usually requires auxiliary information, such as velocity, which is often provided by external equipment. At present, the most commonly used methods are satellite-assisted [1,2,3,4,5,6] and odometer-assisted [7,8,9,10,11] alignment methods. Satellite signals, such as Global Positioning System (GPS) and Beidou, can be easily blocked or interfered with. Thus, they cannot be relied upon during wartime. Hence, the satellite-assisted alignment method has poor anti-interference and autonomy. In contrast, the odometer-assisted inertial navigation system, whether in navigation positioning or initial alignment, has better autonomy and anti-jamming performance. In [12,13], the navigation problem for a land vehicle whose instrumental equipment consists of a strapdown inertial navigation system and an odometer is considered, and a number of very useful functional schemes of solving this problem are discussed in detail, and the specific mathematical description of the corresponding algorithms is presented. In [7,8], high-precision alignment of the moving base was achieved using an odometer-assisted strapdown inertial navigation system without requiring the carrier vehicle to perform a special auxiliary maneuver or stopover. However, if wheel-slip or vehicle-sliding occurs during the odometer-assisted alignment, there will be large errors in the odometer measurement, which can significantly affect the alignment accuracy. In [14,15], the transfer alignment technique for a vehicle-borne master-slave inertial navigation system was investigated, but the technique was difficult to apply in practice due to the difficulties in accurately compensating for the lever arm effect caused by the mounting position of the master-slave inertial navigation system. Reference [16] proposed an initial alignment method for a strapdown inertial navigation system assisted by a Doppler log meter with limited information. In this method, fine alignment was achieved using a nonlinear filter that was based on the rough attitude value.

Based on the Doppler effect, vehicle-borne Doppler radar acquires vehicle speed information using the frequency difference between the emitted electromagnetic waves and the return echoes. This method provides distinct advantages, including high-accuracy, no accumulated errors, continuous output, good anti-interference, and strong autonomy [17,18,19,20]. In addition to normal environmental conditions, Doppler radar can also be applied to rain and snow weather, sand and gravel pavement, and night environment. Its working effect is not affected by the condition of reflective pavement and vehicle sloshing. For this reason, we innovatively proposed that the alignment on the moving base be achieved for the strapdown inertial navigation system assisted by the vehicle-borne Doppler radar, which can effectively avoid measurement errors caused by wheel-slip or vehicle-sliding [18]. We investigated the alignment scheme and the filter algorithm to achieve not only fast and accurate initial alignment on the moving base but also strong interference resistance and high autonomy.

## 2. Moving-Base Alignment Scheme Using a Doppler-Radar-Assisted Strapdown Inertial Navigation System

The east-north-up geographic coordinate system is used as the navigation coordinate system (*n* system). The right-front-up coordinate system is used as the vehicle body coordinate system (*b* system), where up refers to the vertical direction. The strapdown inertial navigation system and Doppler radar are installed on the vehicle in a strap-down manner. The strapdown inertial navigation system provides real-time outputs of the vehicle’s attitude, position, and velocity information in the navigation coordinate system. The Doppler radar provides real-time output of the vehicle’s velocity along the longitudinal axis. Considering that the altitude channel of inertial navigation system is unstable, and the altitude error accumulates rapidly, if the altitude output of inertial navigation system is directly used in the alignment process on moving base, the alignment accuracy will be affected. Therefore, barometric altimeter can be used to assist the inertial navigation system, that is, the altitude output of barometric altimeter can be used to replace the altitude output of the inertial navigation system so as to solve the problem of rapid divergence of the altitude channel of the inertial navigation system.

Before the departure of the vehicle, a quick coarse alignment is first performed on the strapdown inertial navigation system, and a rough value of the vehicle attitude matrix is obtained. The accuracy of the coarse alignment is generally low. Since the interference by the sway of the base and the errors of the inertial device are neglected, the alignment accuracy is usually on the order of 1° [21,22]. After the coarse alignment is complete, vehicle motion is started, and the moving-base alignment begins.

In the process of the moving-base alignment, real-time information of the vehicle’s attitude and velocity is obtained by solving for the dead reckoning using the gyroscopes in the strapdown inertial navigation system and the Doppler radar. In traditional methods, velocity information is usually used only as the measurement of alignment on moving base, but there are some problems such as low accuracy in azimuth alignment and long alignment time. Since both strapdown inertial navigation system and dead reckoning system can output the velocity and attitude information of the vehicle, in addition to the velocity information, the attitude information is introduced into the alignment measurement in this paper. Therefore, the difference between the attitude and velocity output of the strapdown inertial navigation system and the corresponding signal from the dead reckoning system is used as the first measurement. 

In order to further improve the alignment accuracy, more measurement information was introduced in this paper. For ordinary vehicles driven on the road, if the vehicle is not side slipping or jumping, then the transverse and vertical velocities of the vehicle may be viewed as nearly zero. Therefore, the velocity output projection of strapdown inertial navigation system along the transverse and vertical direction of the vehicle body is used as the second measurement innovatively.

Subsequently, the above two measurements are simultaneously fed into the assisted alignment filter, and Kalman filter is used to design the filtering algorithm of assisted alignment. The filtering calculation provides an estimated misalignment angle of the mathematical platform of the strapdown inertial navigation system. This estimate is used to correct the carrier attitude matrix and complete the initial alignment of the strapdown inertial navigation system on the moving base. Thus, the principle of Doppler-radar-assisted alignment of the strapdown inertial navigation system on the moving base is shown in Figure 1.

## 3. Error Model of Gyro/Doppler Radar Dead Reckoning

Since sensor and initial condition errors are always present, the dead reckoning results of the gyro/Doppler radar will always contain errors, including attitude-, velocity-, and position-related errors. Furthermore, these errors accumulate during the iteration process of the solution. Reference [23] has researched the error model of gyro/Doppler radar dead reckoning system in detail. The main error models used in this paper are given here.

### 3.1. Error Model of the Gyroscope and Doppler Radar

Besides errors in the initial conditions, sensor errors are the most significant source of error in the dead reckoning system. This mainly includes gyroscope and Doppler radar errors. For the gyroscope error, what remains after calibration and compensation are mainly constant drift and white noise. Gyroscope errors can generally be written as
(1)$$\varepsilon_{i}=\varepsilon_{bi}+w_{gi},\;i=x,y,z$$
where $\varepsilon_{i}\;(\;i=x,y,z)$ represents the errors of the gyroscope installed along the *x*, *y*, *z* axes of the vehicle body, *w*__BBB__*_gi_*__BBB__ is the white noise of the gyroscope, and $\varepsilon_{bi}$ is the constant drift of the gyroscope that satisfies the following equation:(2)$${\dot{\varepsilon }}_{bi}=0\;(i=x,y,z)$$

Because both the Doppler radar and the strapdown inertial navigation system are strapdown installed on the vehicle, it will inevitably lead to inconsistency between the measurement sensitive axis of the strapdown inertial navigation system and the measurement sensitive axis of the Doppler radar. In general, the installation errors of the strapdown inertial navigation system can be accurately calibrated and compensated beforehand, while some installation errors of Doppler radar will remain. In addition, due to the influence of internal factors and external environment, Doppler radar inevitably has the scale factor error, and it cannot be calibrated accurately in advance. In the process of assisted alignment, the scale factor error and residual installation error of Doppler radar will obviously affect the alignment accuracy, so it is necessary to calibrate and compensate it online. For the strapdown mounted inertial sensors and other auxiliary navigation sensors, it is an effective way to use on-line calibration compensation. At present, it has been widely used in the field of navigation and positioning, and many literatures have also studied it [24,25,26,27,28,29,30,31,32].

The speed measurement error of the Doppler radar is mainly caused by the scale factor error and installation errors [33,34,35,36]. Here, the above errors must be mathematically modeled. The scale factor error is usually viewed as a constant or a first-order Markov process [37,38]. In this paper, we describe its change as a first-order Markov process:(3)$$\delta \dot{k}=-\delta k/\tau_{k}+w_{k}$$
where δk is the scale factor error of the Doppler radar speed measurement, $\tau_{k}$ is the correlation time, and $w_{k}$ is white noise.

The installation errors of the Doppler radar usually can be effectively corrected by periodic calibration [39,40], and then the residual installation errors along three axes are usually viewed as the constants [41], that is
(4)$$\dot{\alpha }=0$$
(5)$$\dot{\beta }=0$$
(6)$$\dot{\gamma }=0$$
where, α,β,γ are the residual installation error angles along the x, y, z axis of the measuring coordinate system, respectively.

### 3.2. System Error Equation in Dead Reckoning

The vehicle attitude matrix, $C_{bD}^{n}$, is obtained by solving the dead reckoning in real-time, and describes the transformation relationship between the vehicle body and navigation coordinate systems. Its function is completely consistent with the attitude matrix of the strapdown inertial navigation system, namely, to realize the function of the so-called “mathematical platform.” Therefore, we may refer to the derivation of the platform attitude error equation of the strapdown inertial navigation system to derive the mathematical platform attitude error equation for the dead reckoning system. This paper shall not dwell on the derivation but instead it will focus on the velocity and position error equations.

Letting the projections of the true and measured values of the vehicle velocity in the measuring coordinate system of Doppler radar be $V_{}^{m}$ and $\hat{V}{}^{m}$, respectively, the presence of the Doppler radar scale factor error, δk, results in the following equation:(7)$$\hat{V}{}^{m}=(1+\delta k)V_{}^{m}$$

Because of the misalignment angle of mathematical platform and installation error angle of Doppler radar in dead reckoning system, the projection of the measured vehicle velocity in the navigation coordinate system is as follows:(8)$$\hat{V}{}^{n}=\hat{C}{}_{bD}^{n}\hat{C}{}_{m}^{bD}\hat{V}{}^{m}$$
where $\hat{C}{}_{bD}^{n}$ is the vehicle attitude matrix, which is obtained by solving the dead reckoning, ${\hat{C}}_{m}^{bD}$ is the actual attitude transfer matrix from the measuring coordinate system to the vehicle body coordinate system.

Letting the mathematical platform misalignment angles of dead reckoning system $\phi_{D}^{n}=$
${\left[\phi_{DE},\phi_{DN},\phi_{DU}\right]}^{T}$, the residual installation error angles of Doppler radar $A={\left[\alpha ,\beta ,\gamma \right]}^{T}$, from Equation (8), we have the following equation:(9)$${\hat{V}}_{}^{n}=(I-\phi_{D}^{n}\times )C_{bD}^{n}(I-A\times )C_{m}^{bD}{\hat{V}}_{}^{m}$$
where $\phi_{DE},\phi_{DN},\phi_{DU}$ are the misalignment angles of the mathematical platform in the east, north, and up directions, respectively.

Substituting Equation (7) into Equation (9) and ignoring second order small errors after expansion, we obtain the following:(10)$${\hat{V}}_{}^{n}=V_{}^{n}-(\phi_{D}^{n}\times )V_{}^{n}+(V_{}^{n}\times )C_{b}^{n}A+\delta kV_{}^{n}$$

From Equation (10), the projection $\delta V_{D}^{n}$ of the velocity error of the dead reckoning in the navigation coordinate system may be expressed as follows:(11)$$\delta V_{D}^{n}=-(\phi_{D}^{n}\times )V_{}^{n}+(V_{}^{n}\times )C_{b}^{n}A+\delta kV_{}^{n}$$

If $\delta V_{D}^{n}$$={\left[\delta v_{DE},\delta v_{DN},\delta v_{DU}\right]}^{T}$ and $V_{}^{n}={\left[v_{E},v_{N},v_{U}\right]}^{T}$, where $\delta v_{DE},\delta v_{DN}$, and $\delta v_{DU}$ are the velocity errors of the dead reckoning in the east, north, and up directions, respectively, we obtain the following equations of the velocity errors by expanding Equation (11):(12)$$\delta v_{DE}=-v_{U}\phi_{DN}+v_{N}\phi_{DU}+(T_{31}v_{N}-T_{21}v_{U})\alpha +(T_{32}v_{N}-T_{22}v_{U})\beta +(T_{33}v_{N}-T_{23}v_{U})\gamma +v_{E}\delta k$$
(13)$$\delta v_{DN}=v_{U}\phi_{DE}-v_{E}\phi_{DU}+(T_{11}v_{U}-T_{31}v_{E})\alpha +(T_{12}v_{U}-T_{32}v_{E})\beta +(T_{13}v_{U}-T_{33}v_{E})\gamma +v_{N}\delta k$$
(14)$$\delta v_{DU}=-v_{N}\phi_{DE}+v_{E}\phi_{DN}+(T_{21}v_{E}-T_{11}v_{N})\alpha +(T_{22}v_{E}-T_{12}v_{N})\beta +(T_{23}v_{E}-T_{13}v_{N})\gamma +v_{U}\delta k$$
where *T_ij_* (*i*, *j* = 1, 2, 3) are the element on the *i*th row and *j*th column of the vehicle’s attitude matrix.

Letting the latitude, longitude, and altitude errors of the dead reckoning system be $\delta L_{D}$, $\delta \lambda_{D}$, and $\delta h_{D}$, respectively, the position error equations based on the geographical coordinate system are as follows:(15)$$\delta {\dot{L}}_{D}=\frac{1}{R_{M}+h}\delta v_{DN}-\frac{v_{N}}{{\left(R_{M}+h\right)}^{2}}\delta h_{D}$$
(16)$$\delta {\dot{\lambda }}_{D}=\frac{\sec L}{R_{N}+h}\delta v_{DE}+\frac{v_{E}\tan L\sec L}{R_{N}+h}\delta L_{D}-\frac{v_{E}\sec L}{{\left(R_{N}+h\right)}^{2}}\delta h_{D}$$
(17)$$\delta {\dot{h}}_{D}=\delta v_{DU}$$

Hence, by substituting Equations (12) to (14) into Equations (15) to (17) and reorganizing, we can obtain the following position error equations for the dead reckoning system:(18)$$\delta {\dot{L}}_{D}=\frac{v_{U}}{R_{M}+h}\phi_{DE}-\frac{v_{E}}{R_{M}+h}\phi_{DU}+\frac{T_{11}v_{U}-T_{31}v_{E}}{R_{M}+h}\alpha +\frac{T_{12}v_{U}-T_{32}v_{E}}{R_{M}+h}\beta +\frac{T_{13}v_{U}-T_{33}v_{E}}{R_{M}+h}\gamma -\frac{v_{N}}{{\left(R_{M}+h\right)}^{2}}\delta h_{D}+\frac{v_{N}}{R_{M}+h}\delta k$$
(19)$$\delta {\dot{\lambda }}_{D}=\frac{\sec L}{R_{N}+h}[v_{N}\phi_{DU}-v_{U}\phi_{DN}+(T_{31}v_{N}-T_{21}v_{U})\alpha +(T_{32}v_{N}-T_{22}v_{U})\beta +(T_{33}v_{N}-T_{23}v_{U})\gamma +v_{E}\delta k]+\;\frac{v_{E}\tan L\sec L}{R_{N}+h}\delta L_{D}-\frac{v_{E}\sec L}{{\left(R_{N}+h\right)}^{2}}\delta h_{D}$$
(20)$$\delta {\dot{h}}_{D}=-v_{N}\phi_{DE}+v_{E}\phi_{DN}+(T_{21}v_{E}-T_{11}v_{N})\alpha +(T_{22}v_{E}-T_{12}v_{N})\beta +(T_{23}v_{E}-T_{13}v_{N})\gamma +v_{U}\delta k$$

## 4. Doppler-Radar-Assisted Moving-Base Alignment Filtering Algorithm for Strapdown Inertial Navigation System

### 4.1. State Equation of Assisted Alignment Filter

The errors of the strapdown inertial navigation system and dead reckoning system are chosen as the state of the assisted alignment filter. The error model of the strapdown inertial navigation system has been reported previously. Errors of the gyroscope/Doppler radar dead reckoning system have been described above. According to Equations (8) to (10), a linear relationship exists between the velocity error of the dead reckoning system, the attitude error of the mathematical platform, and the radar scale factor error, so that the velocity error of the dead reckoning system is no longer included in the state of the assisted alignment filter.

Therefore, the system state of the Doppler radar-assisted moving-base alignment for the strapdown inertial navigation system should include: mathematical platform attitude error of the strapdown inertial navigation system $\phi_{E},\phi_{N},\phi_{U}$, velocity error $\delta v_{E},\delta v_{N},\delta v_{U}$, position error δL,δλ,δh, gyro constant drift $\varepsilon_{bx},\varepsilon_{by},\varepsilon_{bz}$, accelerometer constant bias $\nabla_{bx},\nabla_{by},\nabla_{bz}$, mathematical platform attitude error of the dead reckoning system $\phi_{DE},\phi_{DN},\phi_{DU}$, position error $\delta L_{D},\delta \lambda_{D},\delta h_{D}$, installation error of Doppler radar α,β,γ, and Doppler radar scale factor error δk. Therefore, the system state X of the assisted alignment filter is as follows:(21)$$X={\left[\phi_{E},\phi_{N},\phi_{U},\delta v_{E},\delta v_{N},\delta v_{U},\delta L,\delta \lambda ,\delta h,\varepsilon_{bx},\varepsilon_{by},\varepsilon_{bz},\nabla_{bx},\nabla_{by},\nabla_{bz},\phi_{DE},\phi_{DN},\phi_{DU},\delta L_{D},\delta \lambda_{D},\delta h_{D},\alpha ,\beta ,\gamma ,\delta k\right]}^{T}$$

Hence, based on the error equations of the strapdown inertial navigation system and dead reckoning system, and combined with system state X, the state equation of the Doppler-radar-assisted strapdown inertial navigation system moving-base alignment may be written as follows:(22)$$\dot{X}=FX+GW$$
where F is the system state matrix, G is the system noise driving matrix, and W is the system white noise. Furthermore, $W={\left[w_{gx},w_{gy},w_{gz},w_{ax},w_{ay},w_{az},w_{k}\right]}^{T}$, where $w_{gx},w_{gy}$, and $w_{gz}$ are the gyro white noise; $w_{ax},w_{ay}$, and $w_{az}$ are the accelerometer white noise; and $w_{k}$ is the Doppler radar white noise.

The nonzero terms in matrix F are as follows:$$F(1,2)=\omega_{ie}\sin L+\frac{v_{E}}{R_{N}+h}\tan L,F(1,3)=-(\omega_{ie}\cos L+\frac{v_{E}}{R_{N}+h}),F(1,5)=-\frac{1}{R_{M}+h},F(1,9)=\frac{v_{N}}{{\left(R_{M}+h\right)}^{2}},$$
$$F(2,1)=-\omega_{ie}\sin L-\frac{v_{E}\tan L}{R_{N}+h},F(2,3)=-\frac{v_{N}}{R_{M}+h},F(2,4)=\frac{1}{R_{N}+h},F(2,7)=-\omega_{ie}\sin L,F(2,9)=-\frac{v_{E}}{{\left(R_{N}+h\right)}^{2}},$$
$$F(3,1)=\omega_{ie}\cos L+\frac{v_{E}}{R_{N}+h},F(3,2)=\frac{v_{N}}{R_{M}+h},F(3,4)=\frac{\tan L}{R_{N}+h},F(3,7)=\omega_{ie}\cos L+\frac{v_{E}\sec^{2}L}{R_{N}+h},F(3,9)=-\frac{v_{E}\tan L}{{\left(R_{N}+h\right)}^{2}},$$
$$F(1:3,10:12)=-C_{b}^{n},\;F(4,2)=-f_{U},F(4,3)=f_{N},F(4,4)=\frac{v_{N}\tan L-v_{U}}{R_{N}+h},F(4,5)=2\omega_{ie}\sin L+\frac{v_{E}\tan L}{R_{N}+h},$$
$$F(4,6)=-2\omega_{ie}\cos L-\frac{v_{E}}{R_{N}+h},F(4,7)=2\omega_{ie}(v_{U}\sin L+v_{N}\cos L)+\frac{v_{E}v_{N}}{R_{N}+h}\sec^{2}L,F(4,9)=\frac{v_{E}v_{U}-v_{E}v_{N}\tan L}{{\left(R_{N}+h\right)}^{2}},$$
$$F(5,1)=f_{U},F(5,3)=-f_{E},F(5,4)=-2\omega_{ie}\sin L-\frac{2v_{E}\tan L}{R_{N}+h},F(5,5)=-\frac{v_{U}}{R_{M}+h},F(5,6)=-\frac{v_{N}}{R_{M}+h},$$
$$F(5,7)=-(\frac{v_{E}^{2}\sec^{2}L}{R_{N}+h}+2v_{E}\omega_{ie}\cos L),F(5,9)=\frac{v_{N}v_{U}}{{\left(R_{M}+h\right)}^{2}}+\frac{v_{E}{}^{2}\tan L}{{\left(R_{N}+h\right)}^{2}},\;F(6,1)=-f_{N},F(6,2)=f_{E},$$
$$F(6,4)=2\omega_{ie}\cos L+\frac{2v_{E}}{R_{N}+h},F(6,5)=\frac{2v_{N}}{R_{M}+h},\;F(6,7)=-2v_{E}\omega_{ie}\sin L,F(6,9)=-\frac{v_{E}{}^{2}+v_{N}{}^{2}}{{\left(R_{M}+h\right)}^{2}},$$
$$F(4:6,13:15)=C_{b}^{n},F(7,5)=\frac{1}{R_{M}+h},F(7,9)=-\frac{v_{N}}{{\left(R_{M}+h\right)}^{2}},\;F(8,4)=\frac{\sec L}{R_{N}+h},F(8,7)=\frac{v_{E}\tan L\sec L}{R_{N}+h},$$
$$F(8,9)=-\frac{v_{E}\sec L}{{\left(R_{N}+h\right)}^{2}},F(9,6)=1\;,F(16,16)=-\frac{v_{U}}{R_{M}+h},F(16,17)=\omega_{ie}\sin L+\frac{v_{E}}{R_{N}+h}\tan L,F(16,18)=-\omega_{ie}\cos L,$$
$$F(16,21)=\frac{v_{N}}{{\left(R_{M}+h\right)}^{2}},F(16,22)=\frac{T_{31}v_{E}-T_{11}v_{U}}{R_{M}+h},F(16,23)=\frac{T_{32}v_{E}-T_{12}v_{U}}{R_{M}+h},F(16,24)=\frac{T_{33}v_{E}-T_{13}v_{U}}{R_{M}+h},$$
$$F(16,25)=-\frac{v_{N}}{R_{M}+h},\;F(17,16)=-\omega_{ie}\sin L-\frac{v_{E}\tan L}{R_{N}+h},F(17,19)=-\omega_{ie}\sin L,F(17,17)=-\frac{v_{U}}{R_{N}+h},$$
$$F(17,21)=-\frac{v_{E}}{{\left(R_{N}+h\right)}^{2}},F(17,22)=\frac{T_{31}v_{N}-T_{21}v_{U}}{R_{N}+h},F(17,23)=\frac{T_{32}v_{N}-T_{22}v_{U}}{R_{N}+h},F(17,24)=\frac{T_{33}v_{N}-T_{23}v_{U}}{R_{N}+h},$$
$$F(17,25)=\frac{v_{E}}{R_{N}+h},\;F(18,16)=\omega_{ie}\cos L+\frac{v_{E}}{R_{N}+h},F(18,17)=\frac{v_{N}}{R_{M}+h}-\frac{v_{U}\tan L}{R_{N}+h},F(18,18)=\frac{v_{N}\tan L}{R_{N}+h},$$
$$F(18,19)=\omega_{ie}\cos L+\frac{v_{E}\sec^{2}L}{R_{N}+h},F(18,21)=-\frac{v_{E}\tan L}{{\left(R_{N}+h\right)}^{2}},F(18,22)=\frac{\tan L(T_{31}v_{N}-T_{21}v_{U})}{R_{N}+h},\;F(18,23)=\frac{\tan L(T_{32}v_{N}-T_{22}v_{U})}{R_{N}+h},F(18,24)=\frac{\tan L(T_{33}v_{N}-T_{23}v_{U})}{R_{N}+h},F(18,25)=\frac{v_{E}\tan L}{R_{N}+h},\;F(16:18,10:12)=-C_{b}^{n}$$
$$F(19,16)=\frac{v_{U}}{R_{M}+h},F(19,18)=\frac{-v_{E}}{R_{M}+h},F(19,21)=\frac{-v_{N}}{{\left(R_{M}+h\right)}^{2}},F(19,22)=\frac{T_{11}v_{U}-T_{31}v_{E}}{R_{M}+h},F(19,23)=\frac{T_{12}v_{U}-T_{32}v_{E}}{R_{M}+h},$$
$$F(19,24)=\frac{T_{13}v_{U}-T_{33}v_{E}}{R_{M}+h},F(19,25)=\frac{v_{N}}{R_{M}+h},F(20,17)=-\frac{v_{U}\sec L}{R_{N}+h},F(20,18)=\frac{v_{N}\sec L}{R_{N}+h},$$
$$F(20,19)=\frac{v_{E}\tan L\sec L}{R_{N}+h},F(20,21)=-\frac{v_{E}\sec L}{{\left(R_{N}+h\right)}^{2}},F(20,22)=\frac{(T_{31}v_{N}-T_{21}v_{U})\sec L}{R_{N}+h},$$
$$F(20,23)=\frac{(T_{32}v_{N}-T_{22}v_{U})\sec L}{R_{N}+h},F(20,24)=\frac{(T_{33}v_{N}-T_{23}v_{U})\sec L}{R_{N}+h},F(20,25)=\frac{v_{E}\sec L}{R_{N}+h},$$
$$F(21,16)=-v_{N},F(21,17)=v_{E},F(21,22)=T_{21}v_{E}-T_{11}v_{N},F(21,23)=T_{22}v_{E}-T_{12}v_{N},F(21,24)=T_{23}v_{E}-T_{13}v_{N},$$
$$F(21,25)=v_{U},F(25,25)=-1/\tau_{k}.$$

And the specific form of matrix G is as follows
$$G=\left[\begin{array}{c} \begin{array}{cc} -C_{b}^{n} & O_{3\times 4} \end{array}\\ \begin{array}{cc} O_{3\times 3} & G_{1} \end{array}\\ O_{9\times 7}\\ \begin{array}{cc} -C_{b}^{n} & O_{3\times 4} \end{array}\\ O_{6\times 7}\\ \begin{array}{cc} O_{1\times 3} & G_{2} \end{array} \end{array}\right],\;G_{1}=\left[\begin{array}{cc} C_{b}^{n} & O_{3\times 1} \end{array}\right],\;G_{2}=\left[\begin{array}{cccc} 0 & 0 & 0 & 1 \end{array}\right]$$

### 4.2. Measurement Equation of Assisted Alignment Filter

In traditional methods, the measurement of alignment filter on moving base is usually velocity information. Due to the limited measurement, the alignment accuracy is usually not high, especially in the azimuth direction, and the alignment time is long. Since the gyroscope/Doppler radar dead reckoning system can output the vehicle’s attitude and velocity in real-time, besides velocity information, attitude information is also introduced into the alignment measurement to improve alignment effect. Then, the difference between the attitude and velocity outputs of the strapdown navigation system and dead reckoning system is used as one of the measurements of the assisted alignment filter:(23)$$Z_{1}={\left[\psi_{S}-\psi_{D},\;\theta_{S}-\theta_{D},\;\gamma_{S}-\gamma_{D},\;v_{SE}-v_{DE},\;v_{SN}-v_{DN},\;v_{SU}-v_{DU}\right]}^{T}$$
where $\psi_{S},\theta_{S}$ and $\gamma_{S}$ are the vehicle heading, pitch, and roll angle output of the strapdown inertial navigation system, respectively, $\psi_{D},\theta_{D}$, and $\gamma_{D}$ are the heading, pitch, and roll angle output of the dead reckoning system, respectively, $v_{SE},v_{SN}$, and $v_{SU}$ are the vehicle east, north, and up velocity output of the strapdown inertial navigation system, respectively, and $v_{DE},v_{DN}$, and $v_{DU}$ are the corresponding velocity outputs of the dead reckoning system.

Letting the heading, pitch, and roll angle errors of the strapdown inertial navigation be δψ,δθ, and δγ, respectively, and the heading, pitch, and roll angle errors of dead reckoning be $\delta \psi_{D},\delta \theta_{D}$, and $\delta \gamma_{D}$, respectively, then, according to Equation (23), we have the following:(24)$$Z_{1}={\left[\delta \psi -\delta \psi_{D},\;\delta \theta -\delta \theta_{D},\;\delta \gamma -\delta \gamma_{D},\;\delta v_{E}-\delta v_{DE},\;\delta v_{N}-\delta v_{DN},\;\delta v_{U}-\delta v_{DU}\right]}^{T}$$

Following the derivation in Ref. [5], we have the following:(25)$$\delta \psi -\delta \psi_{D}=-\frac{T_{12}T_{32}}{T_{12}^{2}+T_{22}^{2}}\phi_{E}-\frac{T_{32}T_{22}}{T_{12}^{2}+T_{22}^{2}}\phi_{N}+\phi_{U}+\frac{R_{12}R_{32}}{R_{12}^{2}+R_{22}^{2}}\phi_{DE}+\frac{R_{32}R_{22}}{R_{12}^{2}+R_{22}^{2}}\phi_{DN}-\phi_{DU}$$
(26)$$\delta \theta -\delta \theta_{D}=-\frac{T_{22}}{\sqrt{1-T_{32}^{2}}}\phi_{E}+\frac{T_{12}}{\sqrt{1-T_{32}^{2}}}\phi_{N}+\frac{R_{22}}{\sqrt{1-R_{32}^{2}}}\phi_{DE}-\frac{R_{12}}{\sqrt{1-R_{32}^{2}}}\phi_{DN}$$
(27)$$\delta \gamma -\delta \gamma_{D}=\frac{T_{21}T_{33}-T_{23}T_{31}}{T_{31}^{2}+T_{33}^{2}}\phi_{E}+\frac{T_{13}T_{31}-T_{11}T_{33}}{T_{31}^{2}+T_{33}^{2}}\phi_{N}-\frac{R_{21}R_{33}-R_{23}R_{31}}{R_{31}^{2}+R_{33}^{2}}\phi_{DE}-\frac{R_{13}R_{31}-R_{11}R_{33}}{R_{31}^{2}+R_{33}^{2}}\phi_{DN}$$
where *T_ij_* and *R_ij_* (*i*, *j* = 1, 2, 3) are the element on the *i*th row and *j*th column of the attitude matrix of the strapdown inertial navigation system and dead reckoning system, respectively.

From Equations (12) to (14), we have:(28)$$\delta v_{E}-\delta v_{DE}=\delta v_{E}+v_{U}\phi_{DN}-v_{N}\phi_{DU}-(T_{31}v_{N}-T_{21}v_{U})\alpha -(T_{32}v_{N}-T_{22}v_{U})\beta -(T_{33}v_{N}-T_{23}v_{U})\gamma -v_{E}\delta k$$
(29)$$\delta v_{N}-\delta v_{DN}=\delta v_{N}-v_{U}\phi_{DE}+v_{E}\phi_{DU}-(T_{11}v_{U}-T_{31}v_{E})\alpha -(T_{12}v_{U}-T_{32}v_{E})\beta -(T_{13}v_{U}-T_{33}v_{E})\gamma -v_{N}\delta k$$
(30)$$\delta v_{U}-\delta v_{DU}=\delta v_{U}+v_{N}\phi_{DE}-v_{E}\phi_{DN}-(T_{21}v_{E}-T_{11}v_{N})\alpha -(T_{22}v_{E}-T_{12}v_{N})\beta -(T_{23}v_{E}-T_{13}v_{N})\gamma -v_{U}\delta k$$

One of the measurement equations of the assisted alignment filter can then be obtained by substituting Equations (25) to (30) into Equation (24) and combining the result with the system state ***X***:(31)$$Z_{1}=H_{1}X+V_{1}$$
where ***H***_1_ is the measurement matrix and ***V***_1_ is the measurement white noise series.

To further improve the alignment accuracy, more measurement information is introduced using the constraint conditions on the vehicle motion in this paper. For ordinary vehicles driven on the road, the following motion constraints are satisfied: if the vehicle is not side slipping or jumping, then the transverse and vertical velocities of the vehicle may be viewed as nearly zero [42,43,44]. Based on this motion constraint, the x-axis and z-axis projections in the vehicle body coordinate of the velocity outputs of the strapdown inertial navigation system may be used as the second measurement:(32)$$Z_{2}={\left[{\hat{v}}_{x}^{},{\hat{v}}_{z}^{}\right]}^{T}$$
where ${\hat{v}}_{x}^{}$ and $\hat{v}{}_{z}$ may be calculated from the velocity and attitude matrix outputs of the strapdown inertial navigation system.

Since system errors are always present, the x-axis and z-axis projections of the actual velocity outputs of the strapdown inertial navigation system are not zero but have some errors. Letting the errors be $\delta v_{x}^{}$ and $\delta v_{z}^{}$, according to Equation (32), we have the following:(33)$$Z_{2}={\left[\delta v_{x}^{},\delta v_{z}^{}\right]}^{T}$$

Since the actual vehicle velocity output $\hat{V}{}^{n}$ of the strapdown inertial navigation system in the geographical coordinate system has the following relationship to the projection $\hat{V}{}^{b}$ in the vehicle body coordinate system,
(34)$$\hat{V}{}^{b}=\hat{C}{}_{n}^{b}\hat{V}{}^{n}$$

If we assume that the error in the actual vehicle velocity 
$\hat{V}{}^{n}$ is $\delta V_{}^{n}$ and that the error in the projection $\hat{V}{}^{b}$ of the vehicle velocity is $\delta V_{}^{b}$, according to Equation (34), we obtain the following result:(35)$$V_{}^{b}+\delta V_{}^{b}=C_{n}^{b}(I+[\phi_{}^{n}\times ])(V_{}^{n}+\delta V_{}^{n})$$
where $\left[\phi_{}^{n}\times \right]$ is the cross product anti-symmetric matrix containing the components $\phi_{E},\phi_{N}$, and $\phi_{U}$ of $\phi_{}^{n}$.

Upon expanding the right-hand side of Equation (35) and ignoring the second order small quantities of the error term, we compare the results with the velocity transformation relationship equation $V_{}^{b}=C_{n}^{b}V_{}^{n}$ and obtain the following:(36)$$\delta V_{}^{b}=C_{n}^{b}\delta V_{}^{n}+C_{n}^{b}[\phi_{}^{n}\times ]V_{}^{n}$$

We thereby obtain the following after expanding the right-hand side of Equation (36) and keeping the first and the third rows:(37)$$\left\{ \begin{array}{l} \delta v_{x}^{}=(T_{31}v_{N}-T_{21}v_{U})\phi_{E}+(T_{11}v_{U}-T_{31}v_{E})\phi_{N}+(T_{21}v_{E}-T_{11}v_{N})\phi_{U}+T_{11}\delta v_{E}+T_{21}\delta v_{N}+T_{31}\delta v_{U}\\ \delta v_{z}^{}=(T_{33}v_{N}-T_{23}v_{U})\phi_{E}+(T_{13}v_{U}-T_{33}v_{E})\phi_{N}+(T_{23}v_{E}-T_{13}v_{N})\phi_{U}+T_{13}\delta v_{E}+T_{23}\delta v_{N}+T_{33}\delta v_{U} \end{array}\right.$$

We substitute Equation (37) into Equation (33) and combine the result with the system state ***X*** to obtain the second measurement equation of the assisted alignment:(38)$$Z_{2}=H_{2}X$$
where ***H***_2_ is the measurement matrix.

Together, measurements ***Z***_1_ and ***Z***_2_ are chosen as the measurement of the assisted alignment filter, and the measurement equation for the moving-base alignment of the Doppler-radar-assisted strapdown inertial navigation system is as follows:(39)Z=HX+V
where the measurement vector is $Z=\left[\begin{array}{c} Z_{1}\\ Z_{2} \end{array}\right]$, the measurement matrix is $H=\left[\begin{array}{c} H_{1}\\ H_{2} \end{array}\right]$, and the measurement noise is $V=\left[\begin{array}{c} V_{1}\\ 0_{2\times 1} \end{array}\right]$.

Based on the above state equation and measurement equation for the moving-base alignment, Kalman filter is used to design the filtering algorithm of assisted alignment. Through filtering calculations, the optimal estimates of the misalignment angles $\phi_{E},\phi_{N}$, and $\phi_{U}$ of the mathematical platform of the strapdown inertial navigation system may be deduced. These estimated values are used to correct the attitude matrix $C_{b}^{n}$ of the strapdown inertial navigation system, thereby completing the fine alignment of the strapdown inertial navigation system on a moving base.

## 5. Simulation Validation

In this work, we assumed the following parameter values for the strapdown inertial navigation system: the gyro constant drift was 0.03°/h, with a random walk of 0.003°/h^1/2^; the accelerometer constant bias was 10^−4^ g, with a random walk of 10^−5 PPP^g s^1/2^; the correlation time of the Doppler radar scale factor error was 300 s; and the white noise variance in the velocity measurements was 0.1 m/s; the installation errors of Doppler radar along the *x*, *y*, *z* axes of the vehicle body are 3’, 10’, and 5’, respectively. Before the start of the carrier vehicle, a quick coarse alignment was performed on the strapdown inertial navigation system with a coarse alignment error of 1°, an initial velocity error of 0 m/s, and an initial position error of 15 m. The initial errors of the dead reckoning system were the same as that of the strapdown inertial navigation system. The vehicle was constantly in motion during the alignment process, and the alignment time was taken to be 600 s.

(1) It was assumed that the vehicle started from a stationary position and went through 60 s of linear acceleration motion. After the speed reached 15 m/s, the vehicle continued in a linear motion at a constant velocity until the alignment was complete. Figure 2 shows the trajectory of the vehicle motion.

Based on the motion of Trajectory 1 described above, we conducted simulation validation for cases with and without the introduction of the projection of inertial navigation system velocity output in the vehicle body system as a measurement. The simulation results are shown in Figure 3; Figure 4, where solid lines represent the alignment simulation results with the introduction of the velocity projection as a measurement, and the dashed lines represent the alignment simulation results without introducing the velocity projection measurement.

The results in Figure 3 show that better alignment results were obtained with the radar-assisted strapdown inertial navigation system. Even though the initial coarse alignment had large errors (up to 1°), the estimation errors of the three misalignment angles after fine alignment converged obviously. Good alignment accuracy (with misalignment angle estimation accuracy in the east, north, and up directions better than 0.4’, 0.5’, and 9.8’, respectively) was achieved even without the velocity projection measurement of the inertial navigation system. Furthermore, the simulation results also indicated that the estimation accuracy of the misalignment angle in the up direction was significantly improved; the estimation accuracy was better than 3.2’ when the velocity output projection was introduced into the measurement, and the standard deviation after reaching steady state was 1.79’. However, the effects on the misalignment angle estimation in the two horizontal directions were small: better than 0.5’ in the east direction and better than 0.4’ in the north direction, and the standard deviations after reaching steady state were 0.04’ and 0.02’, respectively. Therefore, we concluded that, based on the vehicle motion constraint, introducing the projection in the vehicle body coordinate system of the inertial navigation system velocity output into the measurement can improve the alignment accuracy on the moving base. At the same time, the results in Figure 4 show that installation errors of Doppler radar along the *x* and *z* axes of the vehicle body were estimated effectively, and the estimated results were very close to the simulation settings (3’ and 5’). However, the installation error along the *y* axis was hardly estimated. 

(2) Figure 5 shows the vehicle motion trajectory when the motion during the alignment process included turning, acceleration, deceleration, uphill motion, and downhill motion in addition to linear motion with constant velocity.

Based on the motion of Trajectory 2 described above, we conducted a simulation validation for the alignment algorithm. The simulation results are shown in Figure 6 and Figure 7, where the solid and dashed lines have the same meaning as described previously.

The results of Figure 6 show that, even when the vehicle underwent turning and acceleration during the alignment process, the method could still maintain high alignment accuracy: with estimation accuracies of the misalignment angles in the east, north, and up directions better than 0.3’, 0.4’, and 3.4’, and the standard deviations after reaching steady state were 0.11’, 0.16’, and 2.19’, respectively. Meanwhile, the influence of vehicle maneuvering on the moving-base alignment was not significant. When the vehicle began to turn at 350 s, the estimation error of the misalignment angle in the up direction increased slowly, but upon completion of the turn, the estimation error gradually decreased again. In addition, we found that during vehicle maneuvering, the alignment accuracy was noticeably better when the inertial navigation velocity projection was included in the measurement than when it was not. Therefore, by analyzing and comparing the simulation results of the two sets of trajectories, we found that the alignment method did not require special maneuvering of the vehicle. On the contrary, various maneuvers should be avoided during the alignment process to facilitate the rapid convergence of the estimation error of the misalignment angle. The results in Figure 7 show that, similarly, the installation error estimations along the *x* and *z* axis of the vehicle were very good, while the installation error estimation along the *y* axis was very poor. The simulation results based on two sets of trajectories show that although the maneuvering forms were different in the course of motion, the estimation effects of Doppler radar installation error along each axis were consistent. 

(3) The moving-base alignment algorithm was simulated assuming that the vehicle experienced multiple side slips and jumps during alignment. For vehicle motion Trajectory 1, we assumed that side slipping occurred at 120, 270, and 450 s, and that jumps occurred at 190, 380, and 540 s, with each event lasting 1.5 s. The simulation results are shown in Figure 8.

The results in Figure 8 show that the moving-base alignment method studied in this work was robust against side slip or jump disturbances of the vehicle. Not only was the alignment not significantly affected, but high alignment accuracy was maintained. The estimation accuracies of the misalignment angles in the east, north, and up directions were better than 0.5’, 0.4’, and 3.5’, and the standard deviations after reaching steady state were 0.19’, 0.23’, and 2.57’, respectively. This is basically equivalent to the alignment accuracy when there was no side slip or jump interference, which further illustrates the superiority of the moving-base alignment for the strapdown inertial navigation system assisted by Doppler radar.

## 6. Conclusions

To summarize, in order to achieve highly accurate and autonomous initial alignment for the vehicle-borne strapdown inertial navigation system under moving-base conditions, we proposed the use of a vehicle-borne Doppler radar to assist a strapdown inertial navigation system for moving-base alignment. The dead reckoning was solved using the gyroscopes of the strapdown inertial navigation system and the Doppler radar. One of the two measurements was the difference between the output attitude and velocity of the strapdown inertial navigation system and the corresponding outputs of the dead reckoning system, and the second measurement was the projection of the output velocity of the strapdown inertial navigation system in the transverse and vertical directions of the vehicle. A filter algorithm was then obtained for assisting the alignment. 

The significance of this study mainly lies in: (1) putting forward the vehicle Doppler Radar to assist strapdown inertial navigation system in motion base alignment, which effectively avoids the measurement errors caused by wheel-slip or vehicle-sliding; (2) besides velocity information, attitude information is introduced into the alignment measurement to improve alignment accuracy and reduce alignment time; (3) in order to further improve the alignment accuracy, more measurement information is introduced by using the vehicle motion constraints, that is, the velocity output projection of strapdown inertial navigation system along the lateral and vertical direction of the vehicle body is also used as the alignment measurement, and the corresponding measurement equation is derived. The experimental results show that the proposed method achieves good alignment effect. Although the coarse alignment error is large, the estimation error of the three misalignment angles after precise alignment converges obviously, and the high alignment accuracy is obtained. In particular, the introduction of the velocity output projection of strapdown inertial navigation system into measurements improves the alignment accuracy on moving base. 

The Doppler radar-assisted moving-base alignment method has advantages, such as high-accuracy, strong autonomy, and good interference resistance. There are no special requirements for the vehicle motion; thus, the method is well-suited for military use and has a high potential for various applications.

## Figures and Tables

**Figure 1 sensors-19-04577-f001:**
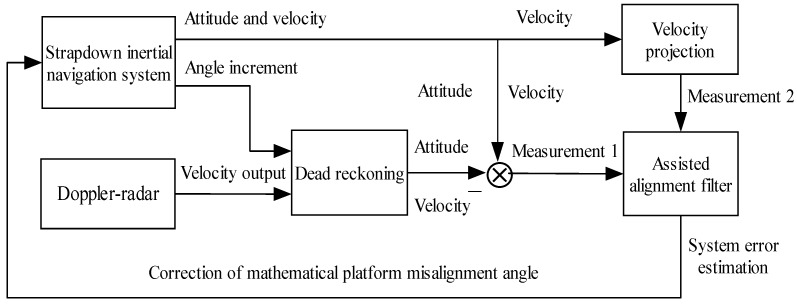
Alignment schematic diagram of strapdown inertial navigation system assisted by Doppler radar on moving base.

**Figure 2 sensors-19-04577-f002:**
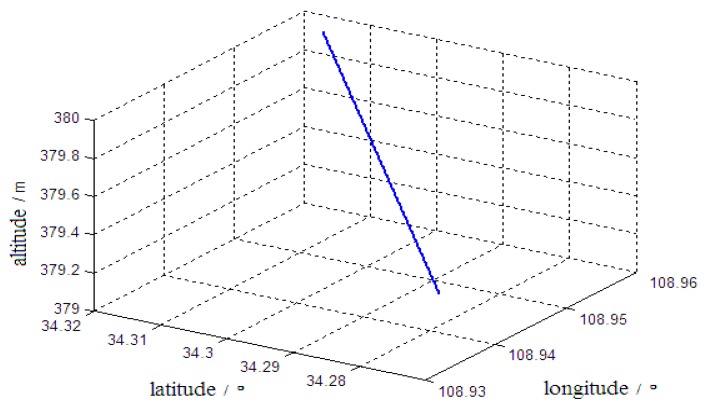
Motion Trajectory 1 of the vehicle.

**Figure 3 sensors-19-04577-f003:**
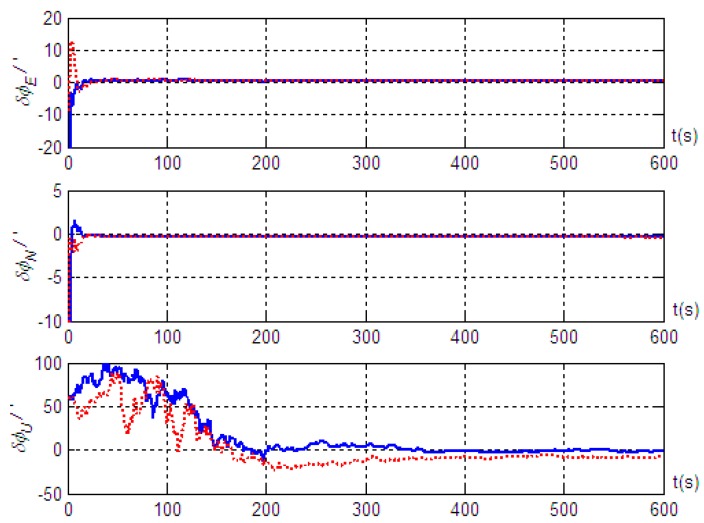
Misalignment angle estimation error of the strapdown inertial navigation system based on motion Trajectory 1.

**Figure 4 sensors-19-04577-f004:**
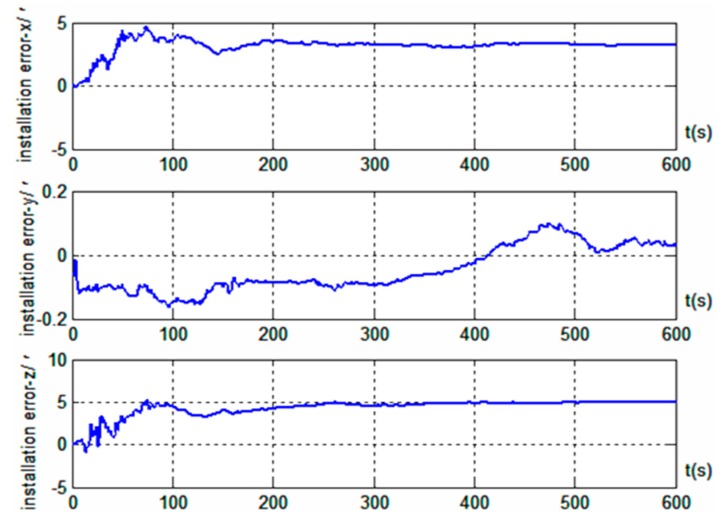
Estimation results of Doppler radar installation error based on motion Trajectory 1.

**Figure 5 sensors-19-04577-f005:**
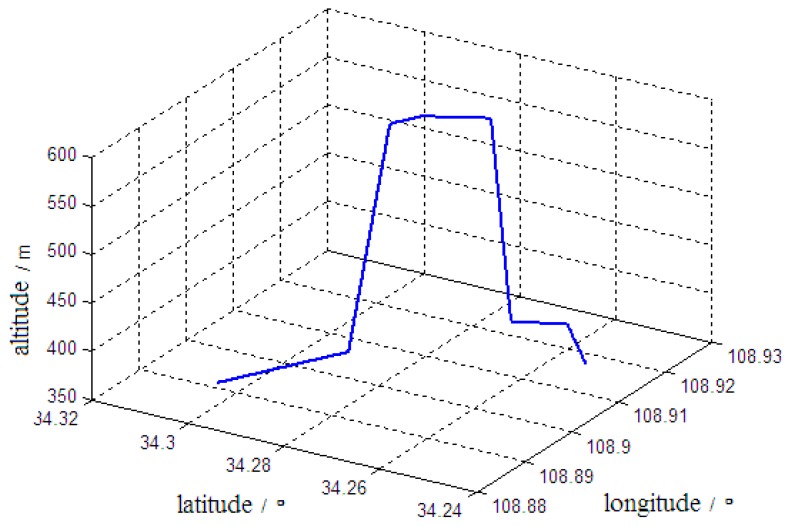
Motion Trajectory 2 of the vehicle.

**Figure 6 sensors-19-04577-f006:**
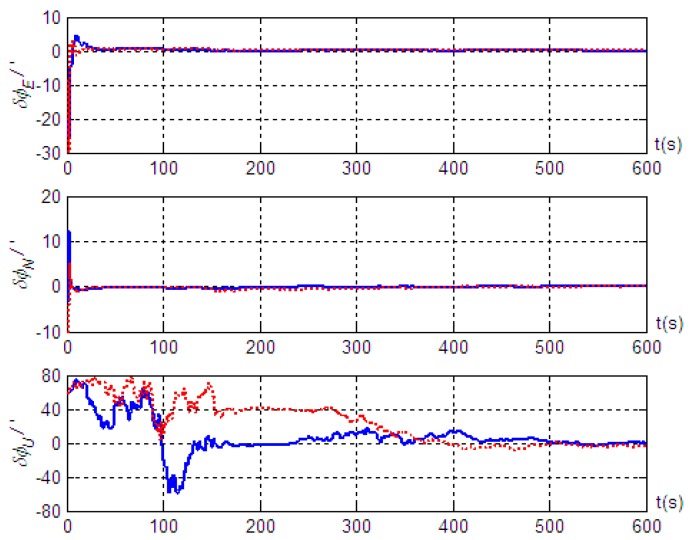
Misalignment angle estimation error of the strapdown inertial navigation system based on motion Trajectory 2.

**Figure 7 sensors-19-04577-f007:**
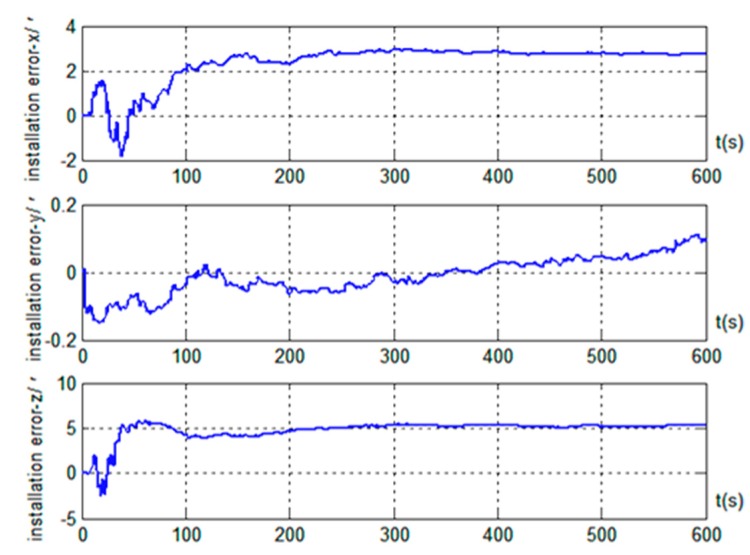
Estimation results of Doppler radar installation error based on motion Trajectory 2.

**Figure 8 sensors-19-04577-f008:**
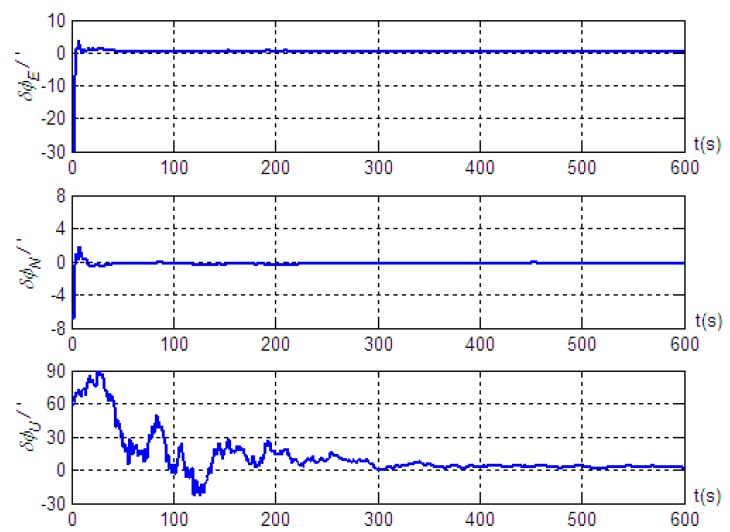
Misalignment angle estimation error of the strapdown inertial navigation system in the presence of side slips and jumps.

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
