# Peer review of "An Alignment Method for Strapdown Inertial Navigation Systems Assisted by Doppler Radar on a Vehicle-Borne Moving Base"

_sensors, 2019, doi:10.3390/s19204577_

Round 1

Reviewer 1 Report

What are the conditions of Doppler radar applicability: when rain is goes, when there are puddles on the ground etc. 2. Since both … Judgment in this phrase is questionable. If the input data for INS mechanization equations and radar aided dead reckoning one uses in optimal manner then one can’t speak about improvement of integrated solution. The usage of internal relations in algorithms with the same input data can not give a qualitative improvement in accuracy. Very close questions were already investigated in: Golovan A.A., Nikitin I.V.·Combined use of strapdown inertial navigation systems and odometers from the standpoint of mechanics of inertial navigation systems. Part 1 DOI: 3103/S0027133015020065, Golovan A.A., Nikitin I.V.·Combined use of strapdown inertial navigation systems and odometers from the standpoint of mechanics of inertial navigation systems. Part 2 DOI: 3103/S0027133015040056 Tip: the residual (after calibration) installation error of the Doppler radar should be included in the state vector of the corresponding estimation problem (see[a],[b]). (19)- (26) are questionable because: The difference in attitude angles values is due to the difference in geographical coordinates (L, l), (LD, lD) only. The exact formulas are presented in [b].

As a consequence, another aiding measurements should be used in mentioned problem instead of (19):

Z = (L – LD, l - lD, h – hD)T

Why V2 is a white noise? In (34) there is no measurement errors at all! 1-Fig.3 present plots of KF estimates. It seems to be more correct to present graphs of the corresponding standard deviations. After reading this article, one may have the illusion that the heading error becomes well observed. It is not so because Doppler radar measurement and property of motion constraint does not affected by motion direction. Achieved estimation accuracy approximately corresponds to accuracy level of so called single gyrocompassing with gyro drifts level of 0.03o/h. It seems necessary to add comment regarding INS mechanization equations of unstable altitude channel. The usage of motion constraint obviously leads to the reduction of Doppler radar vector measurement model (4) (see attached file for formulas)  Vb = (1 + delta*k)V

to the model   Vb = (1 + delta*k)(0, V,0)^T

and eq.(33) ceases to be necessary.

Author Response

Dear Reviewer,

Thank you very much for your comments about our paper submitted to Sensors (Manuscript ID: sensors-592225). These comments are very valuable and helpful for revising and improving our paper, as well as the important guiding significance to our researches.

Now we have carefully checked the manuscript and revised it according to the comments. All the changes in the revised manuscript have been highlighted in yellow. The main corrections in the paper and the responds to the reviewer’ comments are as follows.

Question #1(Reviewer 1):

What are the conditions of Doppler radar applicability: when rain is goes, when there are puddles on the ground etc.

Answer:

In addition to normal environmental conditions, Doppler radar can also be applied to rain and snow weather, sand and gravel pavement and night environment. Its working effect is not affected by the condition of reflective pavement and vehicle sloshing. The above conditions of Doppler radar applicability are supplemented in Section 1 of the revised manuscript.

Question #2(Reviewer 1):

P.2. Since both … Judgment in this phrase is questionable. If the input data for INS mechanization equations and radar aided dead reckoning one uses in optimal manner then one can’t speak about improvement of integrated solution. The usage of internal relations in algorithms with the same input data can not give a qualitative improvement in accuracy.
Answer:

I very much agree with the opinion of the reviewer, so the judgment “to improve alignment accuracy and reduce alignment time” is deleted in the revised version.

Question #3(Reviewer 1):

Very close questions were already investigated in: Golovan A.A., Nikitin I.V.•Combined use of strapdown inertial navigation systems and odometers from the standpoint of mechanics of inertial navigation systems. Part 1 DOI: 3103/S0027133015020065, Golovan A.A., Nikitin I.V.•Combined use of strapdown inertial navigation systems and odometers from the standpoint of mechanics of inertial navigation systems. Part 2 DOI: 3103/S0027133015040056

Answer:

Thank you very much for your reminding. In the above two papers, the navigation problem for a land vehicle whose instrumental equipment consists of a strapdown inertial navigation system and an odometer is considered, and a number of functional schemes of solving this problem are discussed, and the mathematical description of the corresponding algorithms are devoted. These studies are very helpful to our paper, and the research results of these two papers are introduced in Section 1 of the revised manuscript.

Question #4(Reviewer 1):

Tip: the residual (after calibration) installation error of the Doppler radar should be included in the state vector of the corresponding estimation problem (see[a],[b]).

Answer:

In Section 3 and 4 of the revised version, the residual installation error of Doppler radar is considered as random constant, and the error model of gyro/Doppler radar dead reckoning is derived again. The installation error of Doppler radar is included in the state vector of filter estimation, and the measurement equation corresponding to the measurement Z1 is also derived again. On this basis, based on the new state equation and measurement equation, the corresponding simulation results are given in Section 5 of the revised manuscript.

Question #5(Reviewer 1):

Eq (19)- (26) are questionable because: The difference in attitude angles values is due to the difference in geographical coordinates (L,λ), (LD, λD) only. The exact formulas are presented in [b]. As a consequence, another aiding measurements should be used in mentioned problem instead of (19): Z=(L -LD, λ-λD, h -hD)T

Answer:

The problem to be solved in this paper is the initial alignment of strapdown inertial navigation system, that is, to estimate and correct the misalignment angle of strapdown inertial navigation system mathematical platform, rather than to estimate and correct the position error of strapdown inertial navigation system. If the difference of position information is taken as measurement, it is very helpful to estimate the position error of inertial navigation system, but the estimation effect of the misalignment angle of mathematical platform is not as good as that of attitude and velocity information. Because the misalignment angle of the mathematical platform directly affects the attitude and velocity information, and the influence on the position information is indirect. Around this problem, we carry out simulation verification, and the verification results also prove the above conclusions.

Question #6(Reviewer 1):

Why V2 is a white noise? In (34) there is no measurement errors at all.

Answer:

In the revised version, the noise V2 in this equation is deleted.

Question #7(Reviewer 1):

Fig.1-Fig.3 present plots of KF estimates. It seems to be more correct to present graphs of the corresponding standard deviations.

Answer:

In Section 5 of the revised version, the standard deviations corresponding to the Kalman filter estimation results under three kinds of simulation trajectory conditions are given, respectively. Since only one standard deviation value can be obtained by using the estimated result of each filter, only specific values can be given in the revised version, but it is difficult to present graphs of the standard deviations. It is also possible that my understanding is wrong, so I ask the reviewer to further clarify. Thank you very much.

Question #8(Reviewer 1):

After reading this article, one may have the illusion that the heading error becomes well observed. It is not so because Doppler radar measurement and property of motion constraint does not affected by motion direction. Achieved estimation accuracy approximately corresponds to accuracy level of so called single gyrocompassing with gyro drifts level of 0.03o/h.

Answer:

I very much agree with the opinion of the reviewer. The measurement of Doppler radar and motion constraint characteristic are not affected by the motion direction. In this paper, not directly using the output of Doppler radar to construct the measurement of alignment, the attitude and velocity information obtained from the gyro/Doppler radar dead reckoning are used to construct the measurement. Experiments show that, based on the same precision gyroscope and the same initial conditions, the velocity accuracy of dead reckoning system is better than that of strapdown inertial navigation system, and the attitude accuracy is slightly higher than that of strapdown inertial navigation system. Therefore, the results of gyroscope/Doppler radar dead reckoning can effectively assist strapdown inertial navigation system to realize the alignment on moving base. In addition, according to equation (37), for the strapdown inertial navigation system, the x-axis and z-axis projections in the vehicle body coordinate of the velocity output are closely related to the misalignment angles of strapdown inertial navigation system's mathematical platform. Therefore, using the vehicle motion constraints, the projections of the inertial navigation system velocity output along the vehicle body coordinate system are chosen as the measurement, which will improve the alignment accuracy on moving base.

Question #9(Reviewer 1):

It seems necessary to add comment regarding INS mechanization equations of unstable altitude channel.

Answer:

Thank you very much for your reminding. For the inertial navigation system, because of the instability of the altitude channel and the fast accumulation of the altitude error, if the altitude output of the inertial navigation system is directly used in the alignment process, the alignment accuracy will be affected. Therefore, the altitude output of the barometric altimeter can be used instead of the altitude output of the inertial navigation system, which can solve the rapid divergence problem of the altitude channel of the inertial navigation system. In Section 2 of the revised manuscript, this issue is supplemented.

Question #10(Reviewer 1):

The usage of motion constraint obviously leads to the reduction of Doppler radar vector measurement model (please see attached file for formulas) to the model and eq.(33) ceases to be necessary.

Answer:

In this paper, based on the motion constraint characteristic, the velocity output of strapdown inertial navigation system can be used to construct the measurement of the alignment on moving base. It has no direct relationship with Doppler radar, that is to say, if there is no Doppler radar, the motion constraint characteristic can be used to construct the measurement of the alignment in the same way. However, the model (please see attached file for formulas) is the measurement model of Doppler radar, and the use of motion constraints has nothing to do with them. If there is no equation (33), we can not find the relationship between the measurement (please see attached file for formulas) and the system state X , where the measurement Z2 is constructed by the motion constraints. And we can not establish the measurement equation, so equation (33) is necessary.

Thank you again. If you have any question about this paper, please don’t hesitate to let me know.

Reviewer 2 Report

     This paper studies the algorithm of a kind of alignment for SINS assisted by Doppler radar, the algorithm considers both errors of INS and dead reckoning system, and the corresponding differences of motion parameters between two systems and some constraints are introduced into the states of the designed KF, so the alignment precision is raised. The simulation results show the practicality of the proposed alignment method of SINS in a moving base.

      But the innovativeness of this paper is not clear, some algorithms are familiar. The deduced process is correct, but the matrices G and F are not presented in detail, I think it is necessary and very important to readers.

There are two minor problems in this paper:

(1)In equation (21),I doubt the sign of the right side of the equation, please recheck carefully.

(2)Line 245, “where [Φn×] are the cross product anti-symmetric matrices containing the components ΦE ,ΦN , and ΦU of Φn”? [Φn×] is only one matrix, not many matrices.

Author Response

Dear Reviewer,

Thank you very much for your comments about our paper submitted to Sensors (Manuscript ID: sensors-592225). These comments are very valuable and helpful for revising and improving our paper, as well as the important guiding significance to our researches.

Now we have carefully checked the manuscript and revised it according to the comments. All the changes in the revised manuscript have been highlighted in yellow. The main corrections in the paper and the responds to the reviewer’ comments are as follows.

Question #1(Reviewer 2):

But the innovativeness of this paper is not clear, some algorithms are familiar. The deduced process is correct, but the matrices G and F are not presented in detail, I think it is necessary and very important to readers.

Answer:

Thank the reviewer for evaluating my paper. The significance of this study mainly lies in: (1) putting forward the vehicle Doppler Radar to assist strapdown inertial navigation system in motion base alignment, which effectively avoid the measurement errors caused by wheel-slip or vehicle-sliding; (2) besides velocity information, attitude information is introduced into the alignment measurement to improve alignment accuracy and reduce alignment time; (3) In order to further improve the alignment accuracy, more measurement information is introduced by using the vehicle motion constraints, that is, the velocity output projection of strapdown inertial navigation system along the lateral and vertical direction of the vehicle body is also used as the alignment measurement, and the corresponding measurement equation is derived. The experimental results show that the proposed method achieves good alignment effect. Although the coarse alignment error is large, the estimation error of the three misalignment angles after precise alignment converges obviously, and the high alignment accuracy is obtained. In particular, the introduction of the velocity output projection of strapdown inertial navigation system into measurements significantly improves the estimation effect of azimuth misalignment angle, and improves the alignment accuracy on moving base. These works are explained in Section 6 of the revised version. At the same time, the detailed forms of the matrices G and F are given in Section 4 of the revised version.

Question #2(Reviewer 2):

In equation (21),I doubt the sign of the right side of the equation, please recheck carefully.

Answer:

Thank the reviewer for reminding me. I have carefully reviewed this equation and deduced it again, and found that there should be no problem on the right side of the equation.

Question #3(Reviewer 2):

Line 245, “where [Φn×] are the cross product anti-symmetric matrices containing the components ΦE ,ΦN , and ΦU of Φn”? [Φn×] is only one matrix, not many matrices.

Answer:

Thank the reviewer for reminding me. [Φn×] is only one matrix, not many matrices. I'm very sorry for my carelessness. In the revised version, I have changed are to is, and changed materials to matrix.

Thank you again. If you have any question about this paper, please don’t hesitate to let me know.

Reviewer 3 Report

The paper is interesting and treats a problem of practical importance. Many classic references are missing, which are necessary to situate the presented work and its range of application. The problem is related to online calibration which is covered in other references needed inclusion in the present document.

On overall the paper is well written and clear. The weak point is the presentation of the simulation results which deserved an improved exposition. Another point worth supportive evidence is the error model used for the Doppler radar.

List of missing references:

Jurman D. et al. Calibration and data fusion solution for the miniature attitude and heading reference system //Sensors and Actuators A: Physical. – 2007. – Т. 138. – №. 2. – С. 411-420.

Calibration of a MEMS inertial measurement unit, I Skog, P Händel, XVII IMEKO world congress, 1-6

Calibration of the accelerometer triad of an inertial measurement unit, maximum likelihood estimation and Cramer-Rao bound, G Panahandeh, I Skog, M Jansson, 2010 International Conference on Indoor Positioning and Indoor Navigation,

Dorveaux, E., Vissièr, D., & Petit, N. (2010, June). On-the-field calibration of an array of sensors. In Proceedings of the 2010 American Control Conference (pp. 6795-6802). IEEE.   Dorveaux, E., Vissiè, D., Martin, A. P., & Petit, N. (2009, December). Iterative calibration method for inertial and magnetic sensors. In Proceedings of the 48h IEEE Conference on Decision and Control (CDC) held jointly with 2009 28th Chinese Control Conference (pp. 8296-8303). IEEE.

Tedaldi D., Pretto A., Menegatti E. A robust and easy to implement method for IMU calibration without external equipments //2014 IEEE International Conference on Robotics and Automation (ICRA). – IEEE, 2014. – С. 3042-3049.

2] A. B. Chatfield. Fundamentals of High Accuracy Inertial Navigation,
volume 174 of Progress in Astronautics and Aeronautics Seri

[8] D. Gebre-Egziabher, G. Elkaim, J. Powell, and B. Parkinson. A nonlinear,
two-step estimation algorithm for calibrating solid-state strapdown
magnetometers. In 8th International St. Petersburg Conference
on Navigation Systems (IEEE/AIAA), May 200

Author Response

Dear Reviewer,

Thank you very much for your comments about our paper submitted to Sensors (Manuscript ID: sensors-592225). These comments are very valuable and helpful for revising and improving our paper, as well as the important guiding significance to our researches.

Now we have carefully checked the manuscript and revised it according to the comments. All the changes in the revised manuscript have been highlighted in yellow. The main corrections in the paper and the responds to the reviewer’ comments are as follows.

Question #1(Reviewer 3):

The paper is interesting and treats a problem of practical importance. Many classic references are missing, which are necessary to situate the presented work and its range of application. The problem is related to online calibration which is covered in other references needed inclusion in the present document.

Answer:

Thank the reviewer for evaluating my paper. At the same time, thank the reviewer for the reminder and suggestion. In the revised version, I added the relevant classical references, especially those related to online calibration.

Question #2(Reviewer 3):

On overall the paper is well written and clear. The weak point is the presentation of the simulation results which deserved an improved exposition. Another point worth supportive evidence is the error model used for the Doppler radar.

Answer:

Thank you for approving my paper and giving suggestions. In Section 3 of the revised version, in addition to the scale factor error of Doppler radar, the installation error model of Doppler radar is supplemented. The installation errors of Doppler radar are considered as random constants, and the error model of gyro/Doppler radar dead reckoning and the corresponding measurement equation are derived again. At the same time, the references of Doppler radar error model are added. As the reviewer said before, the research work in this paper is closely related to online calibration, so in the simulation verification part of this paper, the online calibration verification results of Doppler radar installation error are supplemented.

Thank you again. If you have any question about this paper, please don’t hesitate to let me know.

Round 2

Reviewer 3 Report

I have no further remarks